# Comparison of the Predicting Performance for Fate of Medial Meniscus Posterior Root Tear Based on Treatment Strategies: A Comparison between Logistic Regression, Gradient Boosting, and CNN Algorithms

**DOI:** 10.3390/diagnostics11071225

**Published:** 2021-07-07

**Authors:** Jae-Ik Lee, Dong-Hyun Kim, Hyun-Jin Yoo, Han-Gyeol Choi, Yong-Seuk Lee

**Affiliations:** Department of Orthopaedic Surgery, Seoul National University College of Medicine, Seoul National University Bundang Hospital, Seongnam 13620, Korea; jaeik15@gmail.com (J.-I.L.); osdrkdh@gmail.com (D.-H.K.); yoo15love@gmail.com (H.-J.Y.); meinmed87@naver.com (H.-G.C.)

**Keywords:** knee, medial meniscus root tear, affecting factors, artificial intelligence, predicting performance

## Abstract

This study aimed to validate the accuracy and prediction performance of machine learning (ML), deep learning (DL), and logistic regression methods in the treatment of medial meniscus posterior root tears (MMPRT). From July 2003 to May 2018, 640 patients diagnosed with MMPRT were included. First, the affecting factors for the surgery were evaluated using statistical analysis. Second, AI technology was introduced using X-ray and MRI. Finally, the accuracy and prediction performance were compared between ML&DL and logistic regression methods. Affecting factors of the logistic regression method corresponded well with the feature importance of the six top-ranked factors in the ML&DL method. There was no significant difference when comparing the accuracy, F1-score, and error rate between ML&DL and logistic regression methods (accuracy = 0.89 and 0.91, F1 score = 0.89 and 0.90, error rate = 0.11 and 0.09; *p* = 0.114, 0.422, and 0.119, respectively). The area under the curve (AUC) values showed excellent test quality for both ML&DL and logistic regression methods (AUC = 0.97 and 0.94, respectively) in the evaluation of prediction performance (*p* = 0.289). The affecting factors of the logistic regression method and the influence of the ML&DL method were not significantly different. The accuracy and performance of the ML&DL method in predicting the fate of MMPRT were comparable to those of the logistic regression method. Therefore, this ML&DL algorithm could potentially predict the outcome of the MMRPT in various fields and situations. Furthermore, our method could be efficiently implemented in current clinical practice.

## 1. Introduction

Osteoarthritis (OA) is an important burden to the healthcare system because it has a higher prevalence with increasing age. The knee joint is the most common site affected by OA. It is a leading cause of disability and has effects on social and public health [1,2]. The prevalence of symptomatic knee OA is estimated to be 38% among Asian adults older than 65 years [3]. The major reason for this high incidence in the Asian population may be related to lifestyle factors such as high flexion activities. In addition, meniscal injury is a known important risk factor that predisposes one to develop OA and is mostly related to degenerative meniscal tears [4]. Medial meniscus posterior root tear (MMPRT) has a high incidence in the Asian population, with increasing concerns about its treatment. The probability of finding an MMPRT in OA patients is estimated at approximately 80%, and MMPRT accounts for 27.8% of all medial meniscal tears [5].

MMPRT has recently received greater attention because it has been associated with the development of excessive meniscal extrusion and accelerated degenerative changes [6]. An important characteristic of MMPRT is that it occurs during the early stages of OA, which implies that the progression of the disease can be prevented or delayed if the MMPRT is properly managed. Therefore, it may be important to identify those patients that are likely to respond well to conservative treatment and those that should be referred for surgery. It would also be helpful to know the affecting factors that determine which treatment strategies to use and the proper screening methods that are prerequisites for successful management.

For the proper treatment of MMPRT, it would be necessary to establish an easy and accurate algorithm rather than rely on the surgeon’s subjective assessment for the standard approach. Nowadays, there has been much interest in using artificial intelligence (AI) in medical imaging techniques such as X-ray and magnetic resonance imaging (MRI) [7]. More than 16,000 peer-reviewed scientific papers in the field of AI are published annually, and orthopedic research comprises a considerable portion of these papers [8]. Using the AI algorithm, it is possible to obtain temporal gain by improving the read efficiency and accuracy of large-scale MRI and X-ray scans. AI, using deep learning (DL) convolutional neural networks (CNN), has recently become a state-of-the-art method for computer vision tasks, such as image classification, with a performance that sometimes surpasses that of humans [9,10]. Characteristics of logistic regression and ML&DL are shown in Table 1.

We conducted this study to identify whether ML&DL is suitable for determining the treatment algorithm for MMPRT similar to a logistic regression method that was previously determined by statistical analysis using affecting factors from the clinical data warehouse (CDW). The AI algorithm was set up with DL and machine learning (ML) using factors that were already analyzed from the CDW. Afterward, the accuracy and predicting performance of ML&DL were compared to those of the logistic regression. The purpose of this study was to validate the accuracy and predicting performance of the ML&DL method by comparing them to those of the logistic regression method. We hypothesized that the ML&DL method will show comparable accuracy and predicting performance with the logistic regression method.

## 2. Materials and Methods

From July 2003 to May 2018, 640 patients who were diagnosed with MMPRT using MRI readings by MSK radiologists or outpatient chart review searching from the CDW in our hospital were included. The exclusion criteria were as follows: (1) a history of trauma such as a periarticular fracture or ligament injury, (2) development of infection, (3) inflammatory arthritis, and (4) previous meniscal injury and/or knee operation. First, affecting factors for the operation were evaluated using statistical analysis. Next, AI technology was introduced using X-ray and MRI machines. Finally, the accuracy and predicting performance were compared between ML&DL and logistic regression methods.

### 2.1. Logistic Regression Method

The logistic regression method refers to a traditional statistical model. Based on previous articles about MMPRT, possible affecting factors of operation or OA progression were surveyed. These factors included age, body mass index (BMI), duration of attempted conservative treatment, malalignment as a weight-bearing line (WBL) ratio, proximal tibial morphology, Kellgren-Lawrence grading scale (K-L grade), bone marrow lesions (BMLs), and severity of meniscal extrusion [11,12,13,14]. Affecting factors were categorized as demographic or radiologic factors. Age, gender, BMI, and conservative interval were included in the demographic category. BMI was defined as the patient’s weight in kilograms divided by the square of height in meters. Conservative interval was determined as an interval between MRI acquisition and an event, which was either the date of operation for patients who underwent such or the date of final outpatient department follow-up for those who participated in conservative treatment.

Radiologic factors were extracted from standing knee anterior–posterior, lateral, hip–knee–ankle (HKA) views, and MRI. WBL ratio, proximal tibial morphology, K-L grade, BMLs, and severity of meniscal extrusion were categorized as radiologic factors. Mechanical axis deviation was evaluated using the initial WBL and delta WBL ratios, while proximal tibial morphology was evaluated using tibial varus angle (TVA) and posterior tibial slope (PTS). Meanwhile, the severity of OA was assessed using the K-L grade and BMLs were evaluated using the MRI osteoarthritis knee score (MOAKS). The extent of medial meniscal extrusion was evaluated on MRI. INFINITT ver. 5.0.9.2 (INFINITT, Seoul, Korea) was used for all radiographic measurements. The WBL ratio was calculated by measuring the distance from the medial edge of the proximal tibia to the point where the WBL intersected the proximal tibia and then by dividing this measurement by the entire width. A percentage was calculated by multiplying this ratio by 100%. The delta WBL ratio was the difference between the initial WBL ratio and the WBL ratio just before the operation or the last follow-up in the outpatient department for patients who underwent conservative treatment. TVA was defined as the angle between the line perpendicular to the tibial shaft and the articular surface of the proximal tibia [15], while PTS was defined as the angle between the line connecting the highest anterior and posterior points of the medial plateau and the line perpendicular to the anterior tibial cortex [11]. The K-L grade was determined using anteroposterior (AP) knee radiographs. Each radiograph was assigned a grade from 0 to 4 based on the extent to which they correlated with increasing OA severity. Grade 0 indicated no presence of OA, and Grade 4 indicated severe OA [16]. The MOAKS instrument refined the scoring of bone marrow lesions (BMLs) [17]. The extent of medial meniscal extrusion was measured from the medial margin of the tibial plateau to the medial margin of the medial meniscus on the image at the midpoint of the femoral condyle [11].

Logistic regression models were applied to binary classification problems, such as the use of surgery or conservative treatment. The characteristics of the logistic regression method were interpretable and derived from coefficients such as odds ratio. The odds ratio was used to determine how each factor influenced the fate of MMPRT. All statistical analyses were performed using the Statistical Package for the Social Sciences (version 22.0, IBM, Armonk, NY, USA). On a priori power analysis, at least 135 patients were required in each group (α = 0.05, β = 0.05).

### 2.2. DL&ML

The dataset of the ML&DL method was shared with that of the logistic regression method. Preprocessing was performed using data cleaning, integration, transformation, reduction, and discretization. Meniscal extrusion and K-L grade were interpreted using the DL method with MRI and X-ray images, as opposed to the logistic regression method that used numerical data. Other numerical data such as age, BMI, conservative interval, TVA, PTS, MOAKS, and WBL ratio were modeled using ML. The results of the DL and ML models were summed using weighted voting and the probability was quantified (Figure 1 and Figure 2). The detailed structures of DL and ML were as follows.

The approach of the DL method was based on the CNN architecture. The detailed network structure for the CNN is summarized in Figure 3. The collected image data consisted of the most visible MRI image cuts for the meniscal extrusion and standing knee AP X-ray images for the K-L grade. The extraction of lesions was performed on the collected image data for model development. Operation and conservative treatment were labeled on the image from which the lesion was extracted. Labeled images were used as input data for the CNN model. The CNN processing pipeline framework was implemented in a hybrid computing environment involving Python (version 3.6; Python Software Foundation, Wilmington, NC, USA) and MATLAB (version 2013; MathWorks, Natick, MA, USA). The CNNs were coded using the Keras package with TensorFlow libraries (version 1.6). The ResNet50 network was used for the extraction of images. The classification CNN in the processing pipeline, which consisted of a 2D ResNet convolutional structure, evaluated structural abnormalities around the joint articular space. Image downsampling was not performed when extracting the image patches. The ResNet50 in the classification CNN was followed by fully connected layers to provide an output probability score within each extracted image patch. Our model predicted a probability distribution for the meniscal extrusion and K-L grades for given images while also highlighting relevant radiological features by generating class-discriminating factors called attention maps. Consequently, we also visualized the attention map, which explained the decision made by the ResNet50 network. For our classifiers, a threefold cross-validation method was applied to randomly classify the 640 patients into 3 equal-sized groups to ensure the independence of the training data from the test data (Table 2).

Similar processes used in DL were also used in ML for preprocessing and data splitting. The boosting series XGBoost model, one of the three models frequently used in structured data analysis, was used for ML. The grid-searching method, which is the process of scanning the data to configure optimal parameters for a given model, was applied for hyperparameter tuning. Before running XGBoost, we set various types of parameters. The learning rate (also called eta and step size shrinkage) was set to 0.3. In this model, the feature weights were reduced to make the boosting process more conservative. Gamma (also called minimum loss reduction) was set to 0. As the maximum depth of the tree increased, the model became more complicated and the possibility of overfitting also increased. Therefore, the max_depth value applied was the default of 6. The subsample was set to 1, which was applied when making a tree (also called iteration), and was also used to prevent overfitting problems. Colsample_bytree, which was a family of parameters for subsampling of columns, was set to 1. As a weight parameter used in the classification model, scale_pos_weight was set to 1. ML was coded by numerical data, excluding meniscal extrusion and K-L grade, which were used differently in DL. The ML model was trained on 80% of the data set using a stratified threefold cross-validation method for initial model tuning, leaving out 20% for final model evaluation.

We performed an analysis to determine those factors that affected the fate of MMPRT. The odds ratio of the logistic regression method and the feature importance of the AI method were compared to assess if they were affecting factors for the fate of MMPRT. The odds ratio was given a statistical significance, while the feature importance was a numerical value of relative influence. Affecting factors of the logistic regression method with statistically significant odds ratios were compared with the rankings of feature importance in the ML&DL method.

Accuracy was compared through a confusion matrix using classification performance evaluation indicators including specificity, recall (also called sensitivity), precision (also called positive predictive value), accuracy, F1 score, and error rate. Among the six parameters, accuracy, F1 score, and error rate were closely related to the accuracy evaluation. Therefore, these parameters were used for the accuracy comparison between logistic regression and ML&DL methods. [18] Predicting performance was evaluated using a receiver operating characteristic curve (ROC). The area under the curve (AUC) was calculated in both logistic regression and ML&DL methods. Finally, accuracy and predicting performance were compared between the two methods.

## 3. Results

The mean age of the patients was 58.09 years (range: 21–86), and the average follow-up period was 3.75 years (range: 2–11) to final treatment. The inter- (kappa = 0.816) and intraobserver (kappa = 0.853) reliabilities were acceptable for assessing X-rays and MRI. Detailed demographic and radiologic data are listed in Table 3.

For the logistic regression method, odds ratios of affecting factors with statistical significance were meniscal extrusion, MRI–Event, initial WBL, delta WBL, K-L grade, and BMI (*p*-values were 0.030, 0.029, 0.042, 0.024, 0.037, and 0.035, respectively). Odds ratios were 4.11, 0.43, 0.67, 7.88, 5.79, and 2.14, respectively. Age, MOAKS, PTS, and TVA showed odds ratios that were not statistically significant: 1.15, 1.22, 1.10, and 0.98, respectively (*p*-values were 0.624, 0.291, 0.442, and 0.735, respectively) (Figure 4A). The confusion matrix was used as part of a method to evaluate the accuracy of the logistic regression model. Classification performance evaluation indicators, which can be calculated using the confusion matrix, were analyzed. These included specificity, recall, precision, accuracy, F1 score, and error rate (0.92, 0.86, 0.92, 0.89, 0.89, and 0.11, respectively) (Figure 5A, Table 4). ROC analyses were conducted to evaluate the performance of the logistic regression method, and the AUC value was measured to be 0.94 for this method (Figure 5B).

Using the feature importance score of the ML&DL method, we investigated which factors affect the fate of MMPRT. The results suggest that the greater the feature importance score was, the more important the affecting factor was to the fate of MMPRT. The feature importance scores of the following parameters were ranked from highest to lowest: meniscal extrusion, MRI–Event, initial WBL, delta WBL, K-L grade, BMI, Age, MOAKS, PTS, and TVA (feature importance scores were 100, 79, 68, 59, 54, 52, 28, 15, 9, and 7, respectively) (Figure 4B). Classification performance evaluation indicators of the ML&DL method were analyzed including specificity, recall, precision, accuracy, F1 score, and error rate (0.98, 0.83, 0.98, 0.91, 0.90, and 0.09, respectively) (Figure 5A, Table 4). AUC values were measured as 0.97 for this method (Figure 5B).

In terms of the comparison of influence (affecting factors) between ML&DL and logistic regression methods, six of the affecting factors that were statistically significant in the logistic regression method also had the highest feature importance scores in the ML&DL method. Classification performance evaluation indicators were as follows: specificities of the logistic regression and ML&DL methods were 0.92 and 0.98, respectively (*p* = 0.010); recalls of the logistic regression and ML&DL methods were 0.86 and 0.83, respectively (*p* = 0.030); precisions of the logistic regression and ML&DL methods were 0.92 and 0.98, respectively (*p* = 0.010). There was no significant difference when comparing the accuracy, F1-score, and error rate between the logistic regression and ML&DL methods (accuracy = 0.89 and 0.91, respectively; F1-score = 0.89 and 0.90, respectively; error rate = 0.11 and 0.09, respectively; *p* = 0.114, 0.422, and 0.119, respectively). The AUC values showed excellent test quality for both ML&DL and logistic regression methods (AUC = 0.97 and 0.94, respectively). There was no significant difference between ML&DL and logistic regression methods in terms of evaluating the predicting performance (*p* = 0.289) (Table 4).

## 4. Discussion

This study compared the accuracy and predicting performance between logistic regression and ML&DL methods. The principal findings of this study include the following: (1) important affecting factors of the logistic regression method for the fate of MMPRT, namely, meniscal extrusion, MRI–Event, initial WBL, delta WBL, K-L grade, and BMI, corresponded well with feature importance of the ML&DL method; (2) the accuracy, F1-score, and error rate of the logistic regression and ML&DL methods were not significantly different; (3) predicting performance was not significantly different between the two groups in terms of the AUC. Therefore, both of our hypotheses were verified.

With regard to evaluating the influence of affecting factors, several studies tried to identify factors that would affect the fate of MMPRT based on treatment strategies. Ford et al. demonstrated significant associations between increasing BMI and meniscal tears, leading to surgical interventions [19]. Hashikawa et al. reported a positive correlation between K-L grade and surgery [20]. Similarly, BMI, meniscal tear, and K-L grade were also important affecting factors in our study. Primeau et al. had reported that varus alignment was consistently identified as a strong predictor of medial knee OA progression in patients with MMPRT [21]. Our study also demonstrated that initial and delta WBL had a strong association with the probability of needing an operation. Meniscal extrusion is another important concern in the progression of OA. Shelburne et al. reported that ≥3-mm medial meniscal extrusion was strongly associated with degenerative joint disease [22]. Another study also reported that meniscal extrusion assessment may be important for determining the optimal treatment strategy for MMPRT [12]. Meniscal extrusion was likewise an important factor in our study. Interestingly, the interval of the conservative treatment, which was the interval between MRI acquisition and an event, was also an important factor for the operation in our study.

Many papers comparing the accuracy and performance of ML&DL and logistic regression methods, a similar area of study as ours, have been published. Accuracy, defined as the proportion of correct predictions, was often used as the result in these papers. However, care had to be taken when using this metric in highly imbalanced datasets. The F1-score was typically used instead of accuracy in cases of severe class imbalance in the dataset. In most papers, accuracy and F1 scores were used together or interchangeably [18]. The DL and ML communities most often used the AUC statistic for comparing model performance [23]. Wang et al. demonstrated that DL with the deep CNN method outperformed the non-DL method for searching prostate cancer [24]. They showed that both accuracy and AUC values of DL were higher than those of the non-DL method. Liu et al. reported that high diagnostic performance was obtained using a fully automated DL-based cartilage lesion detection system [25]. They proved that their DL model had comparable accuracy and diagnostic performance with those of radiologists. Schaffter et al. found that a single ML&DL algorithm could not outperform radiologists for screening mammograms [26]. However, they insisted that an ensemble of ML&DL algorithms combined with radiologist assessment in a single-reader screening environment improved the overall accuracy. Therefore, many studies have reported that the accuracy and performance of the ML&DL method were excellent, compared to those of the logistic regression method. Our results also revealed similar patterns as those seen in previous studies in that the ML&DL method showed comparable accuracy and predicting performance with the logistic regression method.

Our study used DL and ML together to evaluate all important affecting factors. For the meniscal extrusion and K-L grade, convolution features taken from morphologic images of the X-ray and MRI were used to predict the fate of MMPRT. These factors are usually measured or interpreted by orthopedic or radiologic specialists in the clinic. However, our DL model could effectively recognize the convolution features, quantify the meniscal extrusion, and classify the K-L grade. In addition, the probability of fate was calculated by weighted voting using both DL and ML. Therefore, this finding implies that predicting the fate of MMPRT may be possible by obtaining only a few factors from the patient without expert involvement. This enables primary medical facilities to utilize reliable tools in predicting the fate of MMPRT with less involvement from tertiary medical institutions. It can also help to increase the reading efficiency and accuracy of large-scale MRI and X-ray results.

This study has several strengths. First, we tried to establish a treatment algorithm that goes beyond the diagnosis, whereas most studies in the field of ML&DL are limited by focusing on diagnostic accuracy or performance only. It was significant that ML&DL could support clinicians in terms of arriving at a diagnosis and in choosing the best treatment. Second, we used the combined modalities of DL and ML for the evaluation of various kinds of affecting factors. This enabled us to assess all important factors without selection bias. Third, both images and numerical data used in this study were obtained from the CDW of a single institute and were long-term follow-up data. We were able to obtain broad-spectrum information and observe the serial changes in most patients. Finally, our method may be easy to use, and it enables us objectively compare the serial analysis regardless of the location or scale of the hospital.

### Limitations

The current study also has some limitations to be considered. First, our study is limited to the MMRPT in OA that may have various etiologies, and MRI is a prerequisite for establishing the diagnosis of MMPRT. However, MMPRT is an important contributor to OA progression, and it also has a high incidence in the early stages of OA. Second, image data of the WBL was converted to numerical data and was operated in the ML method, which made full automatization impossible although meniscal extrusion and K-L grade used the DL method. It was technically difficult to establish the program to identify the WBL ratio automatically with a whole leg radiograph. Third, the number of data used in our ML&DL model was less than in another ML&DL study. The data numbers were small because we targeted patients with MMPRT as a precondition and not just patients with common OA. However, our dataset included a wide range of information and serial long-term follow-up, which can improve the performance of the ML&DL.

## 5. Conclusions

The affecting factors of the logistic regression method and the influence of the ML&DL method were not significantly different. The accuracy and performance of the ML&DL method in predicting the fate of MMPRT were comparable to those of the logistic regression method. Therefore, this ML&DL algorithm could potentially predict the fate of MMRPT in various fields and situations. Furthermore, our method could be efficiently implemented in current clinical practice.

## Figures and Tables

**Figure 1 diagnostics-11-01225-f001:**
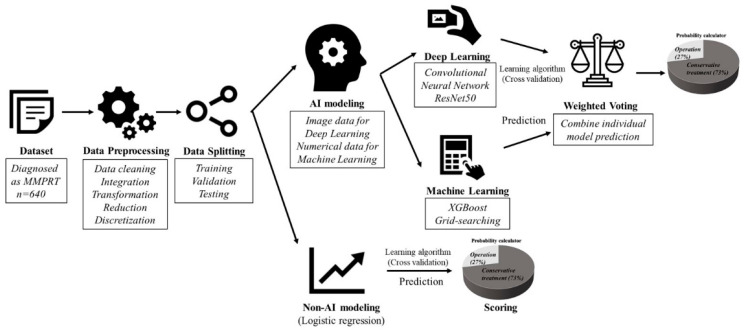
The overall process of ML&DL and logistic regression methods. The process consists of data preprocessing, splitting, modeling, prediction, and probability scoring. The first three steps are similar for both methods while the final two steps differ.

**Figure 2 diagnostics-11-01225-f002:**
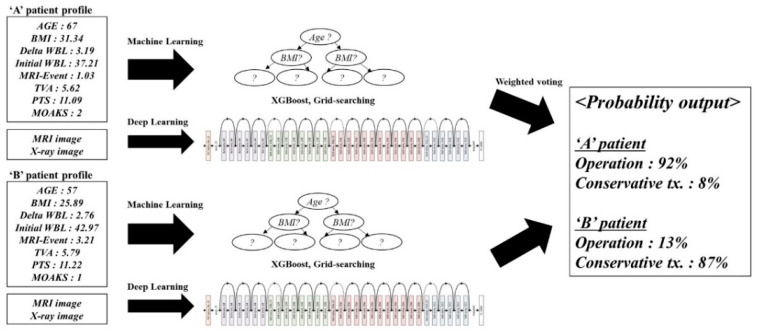
Practical application of the ML&DL method.

**Figure 3 diagnostics-11-01225-f003:**
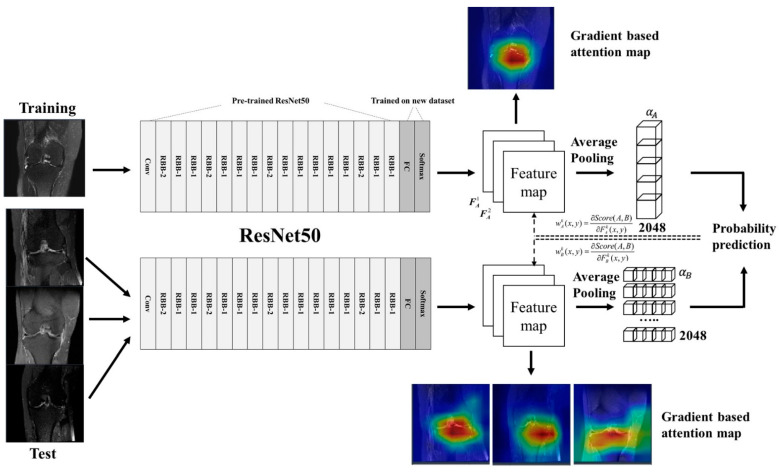
The detailed network structure for the CNNs of DL. Data processing was performed using ResNet50, feature maps were obtained, and probability prediction was calculated using demonstrated formulas.

**Figure 4 diagnostics-11-01225-f004:**
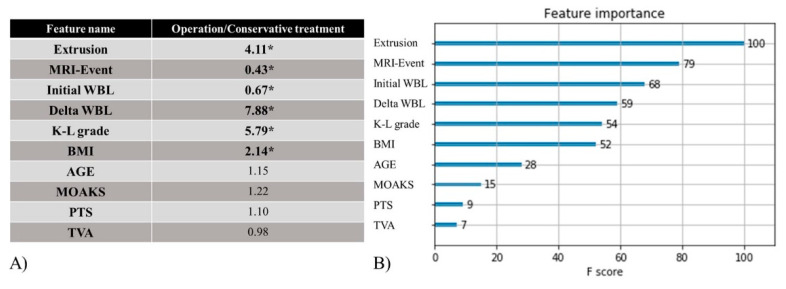
(**A**) Odds ratio of the logistic regression method. * means a significant *p*-value.; (**B**) feature importance of the ML&DL method. Affecting factors of the logistic regression method corresponded well with the feature importance of the six top-ranked factors in the ML&DL method.

**Figure 5 diagnostics-11-01225-f005:**
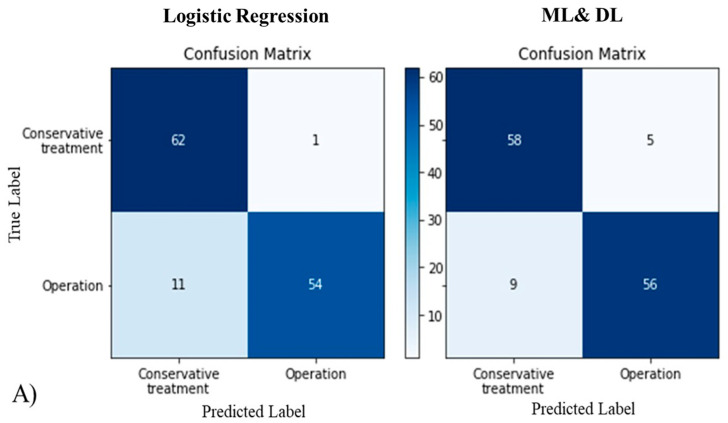
(**A**) As seen in the confusion matrix, there was no significant difference in accuracy when comparing ML&DL and logistic regression methods; (**B**) both methods showed excellent predicting performance with a value greater than 0.9 in the ROC curve.

**Table 1 diagnostics-11-01225-t001:** Differences between logistic regression and ML&DL.

	Logistic Regression	ML and DL
Type of Data	Unstructured or Structured	Standardized in the form
Tools	SPSS	Tensorflow, Python, PyTorch
Applications	Not limited	Not limited
Advantages	It is possible to verify the relationship between the independent variable and the dependent variable.	It is possible to check which factors were important to the accuracy of the model.
Disadvantages	Automation is not possible.	The reliability of the model or the importance of elaborate assumptions is low.

**Table 2 diagnostics-11-01225-t002:** Training and testing data of cross validation.

No. of Group		1	2	3
Training (80%)	All patients	512	510	516
	Operation	201	196	193
	Conservative Tx.	311	314	323
Testing (20%)	All patients	128	130	124
	Operation	40	45	48
	Conservative Tx.	88	85	76

Values are presented as patient numbers.

**Table 3 diagnostics-11-01225-t003:** Affecting factors for the fate of MMPRT in the logistic regression method.

	Operation (*n* = 241)	Conservative Tx. (*n* = 399)
Demographics		
Male/Female	60: 181	155: 244
Age (years)	58.25 ± 20.02	57.13 ± 15.16
BMI (kg/m^2^)	27.71 ± 3.03	25.92 ± 2.87
MRI–Event (years)	1.02 ± 1.12	3.13 ± 2.21
Radiologic factors		
Initial WBL (ratio)	29.41 ± 15.02	36.24 ± 9.32
Delta WBL (ratio)	5.91 ± 1.20	2.76 ± 1.09
TVA (degree)	5.12 ± 1.32	5.07 ± 1.60
PTS (degree)	10.92 ± 2.59	11.11 ± 3.48
K-L grade	3.02 ± 0.88	2.01 ± 1.06
MOAKS	2.14 ± 0.55	1.83 ± 1.02
Extrusion (mm)	3.42 ± 1.24	1.36 ± 1.51

Values are presented as mean ± standard deviation. WBL: weight-bearing line; BMI: body mass index; TVA: tibia varus angle; PTS: posterior tibial slope; MOAKS: MRI osteoarthritis knee score.

**Table 4 diagnostics-11-01225-t004:** Comparison of performance indicators between logistic regression and ML&DL methods.

	Logistic Regression	ML&DL	*p*-Value
Specificity	0.92	0.98	0.010 *
Recall	0.86	0.83	0.030 *
Precision	0.92	0.98	0.010 *
Accuracy	0.89	0.91	0.114
F1-score	0.89	0.90	0.422
Error Rate	0.11	0.09	0.119
AUC	0.94	0.97	0.289

AUC: area under the ROC curve. * means a significant *p*-value.

## Data Availability

The datasets generated during and/or analyzed during the current study are available from the corresponding author on reasonable request.

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
