# Peer review of "Comparison of the Predicting Performance for Fate of Medial Meniscus Posterior Root Tear Based on Treatment Strategies: A Comparison between Logistic Regression, Gradient Boosting, and CNN Algorithms"

_diagnostics, 2021, doi:10.3390/diagnostics11071225_

Round 1
Reviewer 1 Report
The authors have addressed all my queries. I have no more comments to add.
Reviewer 2 Report
The authors have sufficiently replied to the comments of the previous review round.
This manuscript is a resubmission of an earlier submission. The following is a list of the peer review reports and author responses from that submission.
Round 1
Reviewer 1 Report
This is a thorough piece of work. It is well written and well structured. The objective is clear, the methodology well described and the results clearly presented. I have a few comments I believe they could help improve the manuscript.
Major comment: as stated by the authors in the discussion section, the work is based on a small dataset (n= 640 pts from the same hospital) and therefore the risk that the results are not generalizable to other datasets is very high. To avoid this problem, the author should try to validate this methodology on other similar datasets, mostly because the results of their DL/ML system were comparable to those of the non-AI method.
Minor comment: the authors should make an effort to describe how they could improve the DL/ML framework results in the discussion.
Response to Report 1
Major comment: as stated by the authors in the discussion section, the work is based on a small dataset (n= 640 pts from the same hospital) and therefore the risk that the results are not generalizable to other datasets is very high. To avoid this problem, the author should try to validate this methodology on other similar datasets, mostly because the results of their DL/ML system were comparable to those of the non-AI method.
Response: Thank you for your valuable comments. We totally agree with your opinion. It is not possible to generalize everything from a small amount of data. However, the analysis was conducted using minimal numerical data, and no problem was found as a result. The authors will collect and apply additional data later.
Minor comment: the authors should make an effort to describe how they could improve the DL/ML framework results in the discussion.
Response: Thank you for your valuable comments. We focused on parameter tuning, which is thought to have an impact on machine learning and deep learning. To generate the most suitable model for our data, we were able to derive the optimal model by tuning the parameters of each model using the grid search technique.
Reviewer 2 Report
- The paper does not explain clearly its advantages with respect to the literature: it is not clear what is the novelty and contributions of the proposed work: does it propose a new method? Or does the novelty only consist in the application?
- The contributions of the paper are not clearly identified (Section 1, last paragraph). Authors need to be claimed their contributions and justify with sufficient experimental results.
- I recommend adding a new section for the related work. In addition, I recommend adding a Table for related works and show the advantages and disadvantages of each study.
- Analysis parts need to be well written as take-home-messages are not clear. For example, one paragraph - one message.
- Authors should add more details about the data.
- Authors should revise the references in the text as it appears before the text (e.g. ref [1,2].
- I’m wondering if the authors can provide the following information to ensure reproducibility of the work: (1) A clear description of the mathematical setting, algorithm, and/or model; (2) A link to a downloadable source code, with the specification of all dependencies, including external libraries; (3) The average runtime for the algorithm, or estimated energy cost; The number of parameters in the model; (4) A clear definition of the specific evaluation measure or statistics used to report results; (5) The exact number of training and evaluation runs.
- A number of very brief paragraphs- they are confusing and non-informative.
- The paper would benefit much if you ask a native speaker to review and edit the text with a focus on the usage of the English language.
Response to Report 2
The paper does not explain clearly its advantages with respect to the literature: it is not clear what is the novelty and contributions of the proposed work: does it propose a new method? Or does the novelty only consist in the application?
Response: Thank you for your valuable comments. The authors compared the conventional method with the modern method using ML/DL. These comparisons revealed no difference. We are not suggesting a new method. We compared the two existing methods and thought about which method is a little more convenient. As a result, the authors suggested a new application that can be used more easily in a more convenient way.
The contributions of the paper are not clearly identified (Section 1, last paragraph). Authors need to be claimed their contributions and justify with sufficient experimental results.
Response: Thank you for your valuable comments. Contributions of all authors are listed after the conclusion paragraph. (Line 351-354)
I recommend adding a new section for the related work. In addition, I recommend adding a Table for related works and show the advantages and disadvantages of each study.
Response: Thank you for your valuable comments. A table comparing statistical analysis and ML/DL has been added to the introduction section. Table 1 (Line 62)
Analysis parts need to be well written as take-home-messages are not clear. For example, one paragraph - one message.
Response: Thank you for your valuable comments. We fully agree with your point. Methodologically, if we could make it a little simpler when comparing the two methods, we would have written take home message in an easy-to-understand way. However, in the conclusion section we clearly state the purpose, rationale, and utility of our study.
Authors should add more details about the data.
Response: Thank you for your valuable comments. All data was taken from CDW of SNUBH. Which patients were included and which patients were excluded are noted in the first paragraph of the materials and methods section.
Authors should revise the references in the text as it appears before the text (e.g. ref [1,2]. 결과보고에 사용되는 특정 평가 조치 또는 통계에 대한 명확한 정의
Response: Thank you for your valuable comments. Corrected the format of all references.
I’m wondering if the authors can provide the following information to ensure reproducibility of the work: (1) A clear description of the mathematical setting, algorithm, and/or model; (2) A link to a downloadable source code, with the specification of all dependencies, including external libraries; (3) The average runtime for the algorithm, or estimated energy cost; The number of parameters in the model; (4) A clear definition of the specific evaluation measure or statistics used to report results; (5) The exact number of training and evaluation runs.
Response: Thank you for your valuable comments.
(1) Logistic regression was used for statistical analysis, and Gradient Boosting and CNN Algorithms were used for ML and DL.
(2) The source code is attached as a file. The package list is as follows.
pandas, seaborn, matploltlib, tensorflow, keras, sklearn
(3) The average time required was around 3-5 minutes. For ML and DL, the number of parameters was 5.
(4) The classification result table (Confusion Matrix) is a table or display of the results counting whether the original class of the target matches the class predicted by the model
(5) A total of 10 learning times were performed.
A number of very brief paragraphs- they are confusing and non-informative.
Response: Thank you for your valuable comments. If you point out which part is the problem, we will fix it. Most meaningless paragraphs have been corrected.
The paper would benefit much if you ask a native speaker to review and edit the text with a focus on the usage of the English language.
Response: Thank you for your valuable comments. It has been re-corrected twice by a native speaker.
Reviewer 3 Report
I read with interest this study on the comparison of logistic regression, XGBoost and a CNN for the diagnosis of root tears. The work has important methodological flaws. Specifically: 1. The authors devote a whole first paragraph talking about the relationship between OA and meniscal root tears. However, the first thing that needs to be discussed is the relationship between trauma and root tears. Please start by discussing trauma and mechanisms causing root tears and then move to the development of OA as a result of root tears. 2. The text needs extensive language editing since it is full of syntactical and grammatical inaccuracies. 3. the idea that you are comparing AI vs non-AI is not correct. Unfortunately AI is a general term that covers both primitive methods such as logistic regression and methods such as DL and ML. Therefore your title and the whole text needs to be modified to reflect this concept. Instead of talking about AI vs non-AI you can talk about regression vs ML and DL. This needs to be changed in the title and everywhere in the text. For example the second part of the title could be changed to ": a comparison between logistic regression, gradient boosting and CNN algorithms" 4. In order to ensure reproducibility of your model it would be good if the ML and DL code were added as a supplementary or uploaded in GitHub. 5. line 181 should be a title 6. For the logistic regression model please present the table with odds ratios and p-values that SPSS gives you when you run logistic regression. please also give a p-value for the whole logistic regression model. It is important to know whether the factors are independent predictors of the outcome. 7. it is unclear how you converted the images to data to run the XGboost algorithm. Did you segment the tears? if not, segmentation is important. 8. how did you normalise the values between images for the XGboost? 9. In figure 2 it is impossible to see what represents your CNN method. the size needs to be increased. 10. Who performed the labelling? who diagnosed the tears? how many raters were used? were they experienced in MSK radiology? all these questions need to be answered in the text since the diagnosis of root tears is not trivial and cannot be done with the same accuracy between a radiology resident, a general radiologist and an MSK radiologist. Orthopaedic surgeons do not have the proper training to diagnose meniscal tears on MRI. 11. line 325: which other AI research? 12. please perform a power calculation for your logistic regression. 13. You need to provide more papers doing meniscal tear diagnosis with AI and discuss them all comparing to your results. 14. It is not clear why you excluded patients with trauma since you are examining root tears. Root tears resulting from OA are a minority.
Response to Report 3
- The authors devote a whole first paragraph talking about the relationship between OA and meniscal root tears. However, the first thing that needs to be discussed is the relationship between trauma and root tears. Please start by discussing trauma and mechanisms causing root tears and then move to the development of OA as a result of root tears.
Response: Thank you for your valuable comments. The authors studied degenerative root tears, excluding traumatic root tears. Traumatic patients are more likely to be accompanied by other lesions. In addition, it is difficult to infer the contribution of the affecting factors to surgery.
- The text needs extensive language editing since it is full of syntactical and grammatical inaccuracies.
Response: Thank you for your valuable comments. It has been re-corrected twice by a native speaker.
- the idea that you are comparing AI vs non-AI is not correct. Unfortunately AI is a general term that covers both primitive methods such as logistic regression and methods such as DL and ML. Therefore your title and the whole text needs to be modified to reflect this concept. Instead of talking about AI vs non-AI you can talk about regression vs ML and DL. This needs to be changed in the title and everywhere in the text. For example the second part of the title could be changed to ": a comparison between logistic regression, gradient boosting and CNN algorithms"
Response: Thank you for your valuable comments. We have corrected all the parts as you advised.
- In order to ensure reproducibility of your model it would be good if the ML and DL code were added as a supplementary or uploaded in GitHub.
Response: Thank you for your valuable comments. The source code is attached as a file.
- line 181 should be a title
Response: Thank you for your valuable comments. That part has been deleted.
- For the logistic regression model please present the table with odds ratios and p-values that SPSS gives you when you run logistic regression. please also give a p-value for the whole logistic regression model. It is important to know whether the factors are independent predictors of the outcome.
Response: Thank you for your valuable comments. The odds ratio is presented in figure 4. Also, the p-value for each is written in the result section. (Line 212-217, 224)
- it is unclear how you converted the images to data to run the XGboost algorithm. Did you segment the tears? if not, segmentation is important.
Response: Thank you for your valuable comments. Segmentation was not performed, and the data was compressed in one dimension in a line that did not damage the characteristics of the image data as much as possible.
- how did you normalise the values between images for the XGboost?
Response: Thank you for your valuable comments. Before changing to one-dimensional, the pixel value of the image data was divided by 255 and converted to 0~1, and then it was changed to one-dimensional compressed data.
- In figure 2 it is impossible to see what represents your CNN method. the size needs to be increased.
Response: Thank you for your valuable comments. Since resnet50 is a well-known model, we decided that it is not necessary to show the detailed structure. Rather than the detailed structure of resnet50, how the probability was calculated is more important.
- Who performed the labelling? who diagnosed the tears? how many raters were used? were they experienced in MSK radiology? all these questions need to be answered in the text since the diagnosis of root tears is not trivial and cannot be done with the same accuracy between a radiology resident, a general radiologist and an MSK radiologist. Orthopaedic surgeons do not have the proper training to diagnose meniscal tears on MRI.
Response: Thank you for your valuable comments. All root tears were diagnosed by a musculoskeletal radiologist. Cases with root tears in MRI readings diagnosed by radiologists were included in the patient group. Edited to include a root tear diagnosed by MSK radiologist in the text.
- line 325: which other AI research?
Response: Thank you for your valuable comments. It refers to the general AI research, and it was usually targeted to a group of more than 1000 cases.
- please perform a power calculation for your logistic regression.
Response: Thank you for your valuable comments. On a priori power analysis, at least 135 patients were required in each group (α = 0.05, β = 0.05).
- You need to provide more papers doing meniscal tear diagnosis with AI and discuss them all comparing to your results.
Response: Thank you for your valuable comments. A paper on deep learning to detect abnormalities in knee MRI has already been published. Our paper is not about AI diagnosing diseases using MRI. Patients who have already been diagnosed by a musculoskeletal radiologist were targeted. MRI and x-ray were used as means to input affecting factors into DL.
- It is not clear why you excluded patients with trauma since you are examining root tears. Root tears resulting from OA are a minority.
Response: Thank you for your valuable comments. Traumatic root tears in young patient and degenerative root tears in older patient are completely different diseases. The purpose of our paper is to predict the future of any patient by analyzing the affecting factors. However, in the case of trauma patients, it is difficult to predict the future because most of these affecting factors are related to degeneration. In addition, since trauma patients may have lesions other than root tears, only pure degenerative root tears were included in the study.